# Experimental Study on Pore Pressure Variation and Erosion Stability of Sandy Slope Model under Microbially Induced Carbonate Precipitation

**Mingjuan Huang [1], Youliang Zhang [2,*], Jinning Hu [2], Yunpeng Hei [2], Zikun Xu [1] and Jinchen Su [1]**

[1] Hainan Provincial Water Conservancy and Hydropower Group Co., Ltd., Haikou 571126, China; huangmingjuan2023@163.com (M.H.); xuzikun2023@163.com (Z.X.); sujinchen2023@163.com (J.S.)

[2] School of Civil and Architectural Engineering, Hainan University, Haikou 570228, China; 951508049@163.com (J.H.); 21220856000152@hainanu.edu.cn (Y.H.)

[*] Correspondence: zhangyouliang@hainanu.edu.cn

**Abstract:** With the development of a free trade port on Hainan Island, the construction of tourist roads around the island is currently underway. However, the weather conditions on Hainan Island, which include strong typhoons and rainstorms, pose challenges for the construction of highway-cutting slopes on the coastal weak sandy terraces. These slopes are susceptible to sand loss and erosion from rainfall. To address this issue, MICP green spray irrigation solidification technology is used to strengthen the sandy cutting, and pore water pressure monitoring is carried out on the slope model during MICP solidification and rainfall scour. Combined with the model pore water pressure and flow slip failure pattern, a dynamic analysis was conducted. The results show that MICP sprinkler irrigation technology can solidify the surface of the slope model in a short time, and after three sets of rotation reinforcement, the model achieved a cementation depth of 4 cm, with a well-reinforced surface and closely connected sand samples. Under the erosion effect of simulated rainfall intensity, the sand loss of the slope was weakened, without damage to the sand binding, and the integrity was enhanced. The cementation between the sand grains facilitated the conversion of most of the rainfall into runoff. However, despite these efforts, the slope eventually slid after 150 s. During the sliding process, the leading edge of the slope model lost sand and became unloaded, and the failure mode was graded a creep slip failure. Finally, the slope was divided into several blocks due to the continuous expansion of cracks following the slope failure. The erosion stability of the sandy slope under heavy rains was optimized and the sand loss was prevented effectively. This study proposes a new method of MICP remediation techniques that serve as a new test basis for the practical application of MICP technology in engineering projects.

**Keywords:** MICP; sandy slope; pore water pressure; fluid slip pattern

## 1. Introduction

In most projects located in tropical areas, many soil slopes are distributed under natural conditions or for engineering purposes. Engineering measures can be used to strengthen the soil slope, among which the use of microbial grouting to improve and to enhance soil slope stability holds significant engineering application value. At the same time, abundant rain and typhoons often lead to the unsaturated state of the slope soil, and continuous heavy rain causes rainwater to infiltrate the slope soil. This results in increased saturation of the slope soil and reduced matric suction of unsaturated soil. Consequently, there is a substantial reduction in the shear strength of the slope soil and a surge in the possibility of landslide disasters. Therefore, it is of great theoretical and engineering value to study the stability of a slope reinforced by microbial grouting under the condition of rainfall infiltration.

The technology of microbially induced carbonate precipitation (MICP) utilizes the metabolic reactions of the Bacillus bacteria and the nutrients present in the culture environment to promote a series of chemical reactions. This bacterium is abundant in the soil, well-adapted to the environment and decomposes man-made urea at an extremely fast rate. Nutrients are absorbed through the nitrification and decomposition reaction and converted into products mainly composed of carbonate. These products gather free ions to form large complex groups, which exhibit a strong cementation effect and adsorb the generated calcium carbonate crystals around the sclerotium. At the same time, the continuous production of $CO_3^{2-}$ ions from decomposition is transported to the surface of the bacteria and combined with $Ca^{2+}$ ions adsorbed on the outer surface to precipitate calcium carbonate. Through these calcium carbonate "Bridges", the surrounding loose particles can be condensed into a structurally robust whole, which can fill the pores of the sand body [1,2]. The main biochemical equation of calcium carbonate precipitation induced by microorganisms is as follows [3]:

$$Cell + Ca^{2+} \longrightarrow Cell\text{-}Ca^{2+} \tag{1}$$

$$NH_2\text{-}CO\text{-}NH_2 + 2H_2O \longrightarrow 2NH_4^+ + CO_3^{2-} + Cell + Ca^{2+} \tag{2}$$

$$CO_3^{2-} + Cell\text{-}Ca^{2+} \longrightarrow Cell\text{-}CaCO_3 \downarrow \tag{3}$$

Based on this reaction mechanism, many scholars at home and abroad have conducted in-depth studies on grouting materials, methods, liquefaction and other aspects [3]. Shao Guanghui et al. [4] found that bacterial biomass was the fundamental factor affecting microbial solidification, with a linear decrease in the bacterial content along the reinforcement channel and a gradual decline in activity. Li Xian et al. [5], based on soil permeability characteristics and on the premise of the compatibility of microbial survival, conducted tests on the applicability of MICP sand for several soils. They concluded that the reaction environment of MICP hydrolysis and the induced formation of calcium carbonate is liquid, making it more suitable for sand soil. Xu Pengxu et al. [6] performed single-channel reinforcement with a peristaltic pump on sand samples of different particle sizes. They concluded that if the pores of the sand samples were too small, the upper part of the sand samples would be prone to blockage, leading to an uneven reinforcement effect. On the other hand, if the particle size was too large, the cementation effect would not be ideal, because the liquid could not easily remain, resulting in a smaller microbial contact surface. In another study, Whiffin [7] performed a long sand column test lasting 5 days and set the consolidation drainage shear test under 50 kPa confining pressure. After the test, the strength of the sand column reached 200 kPa~570 kPa. Ma Ruinan et al. [8] used the mixing method to explore the permeability characteristics of calcareous sand. They found that MICP technology can reduce the permeability coefficient of calcareous sand by 1 to 2 orders of magnitude, improve the adhesion of calcareous sand and prevent the sand skeleton from being easily eroded by seepage. Liu et al. [9] conducted a relevant experimental study on MICP-reinforced calcareous sand. Their results showed that an increase in calcium carbonate content would not affect the peak internal friction angle of the reinforced soil but increased the cohesion of the soil.

In view of the mechanical properties of MICP-reinforced sand, Fang et al. [10,11] studied the influences of different confining pressures and cementation degrees on the mechanical properties of quartz sand through triaxial drainage tests. The results demonstrate that the strength, dilatancy and initial elastic modulus of MICP-reinforced sand are directly proportional to the degree of cementation, and the degree of cementation is found to be closely linked to the bonding force. Yiman Bing et al. [12,13] conducted experiments that measured the shear strength, compressibility and other properties of the MICP-reinforced coastal soft soil. The outcomes indicated that MICP technology could improve the compression characteristics of the soft soil and improve the shear strength of

the reshaped soil. Tang Yang proposed the parameter-sensitive sequence and combination scheme for MICP-reinforced soft soil. Yin Liyang et al. [14,15] conducted research to verify the correlation between soil strength and calcium carbonate generation. They found that the content of calcium carbonate represented the overall level of cementation in the soil.

At present, there is limited research on the applicability of MICP-reinforced slopes under rainfall erosion and infiltration conditions. Rainfall infiltration is a major factor for an increase in the water content of slopes and a decrease in the shear strength of soil on slopes, leading to a decline in the slope under stability. As the water content rises, the anti-sliding force of the slope decreases. Hu Qizhi et al. [16] used MIDAS/GTS software to analyze and calculate the stability of slopes under fluid–structure coupling under the influence of mattress suction effects. They proved that microbial reinforcement mitigates the influence of rainfall infiltration on slopes, reduces the negative pressure area of pore water pressure on the slope and improves the stability of the slope under rainfall conditions. Wang Zhaoyang et al. [17] conducted tests where they changed the water level in the model to reveal the correlation between the changes in the groundwater level and the pore water in the soil. They observed that the negative pressure of unsaturated soil increases under the action of rainfall and loading, indicating a relationship between the change in pore water pressure and the water content in the soil. Zeng Qiang [18] conducted a comparative analysis of influencing factors such as pore water pressure, volume water content and the shear stress of the slope under rainfall conditions. Their findings revealed that with increasing rainfall intensity under constant rainfall duration, the changes in the shear strain and displacement of the slope surface also increase, and the shear strain of the slope soil gradually moves inwards and backwards, especially at the foot of the slope where the stress concentration is obvious. The increase in shear strain will cause slope instability.

At present, both domestic and international scholars have conducted in-depth research on MICP-reinforced sand bodies, but there is still a lack of research on coastal fine sand, and the study on solidified marine fine sand is limited. In the existing research results, most of the reinforcement methods adopted in the study include single-channel grouting reinforcement, mixing reinforcement and direct irrigation, and there is little research on sprinkler irrigation reinforcement. This method has the advantage of maximizing the contact between bacterial liquid and air promoting the vaporization of the bacterial liquid and increasing the contact area with sand samples. As a result, it induces the generation of a large number of calcium carbonate crystals and improves the reinforcement effect. There are few studies on the sandy slope reinforced by MICP under the action of rainfall. Therefore, it is necessary to further analyze the related factors that affect the reinforcement effect of the water solubility of calcium carbonate generated after reinforcement when the water content is high. Additionally, it is essential to study particle connectivity under high pore water pressure conditions.

Based on microbial grouting to strengthen the slope, this study applied the microbial mortar generated by microbial grouting for the slope reinforcement. Grouting tests were conducted to obtain valuable data on microbial grouting. The comparison was made between the pore water pressure parameters and flow slip morphology of soil samples after grouting. Based on these comparisons, it is comprehensively evaluated that the microbial mortar slope after grouting exhibited improved water stability and erosion resistance under the action of heavy rainfall, which improves the safety factor of the slope.

## 2. Test Materials and Devices

### 2.1. Test Sand

The medium-fine sand used in the test was extracted from the actual engineering site with standard roundness. Adjacent to the site were new road cuts, containing pical beach terrace sand. The engineering environment is shown in the Figure 1.

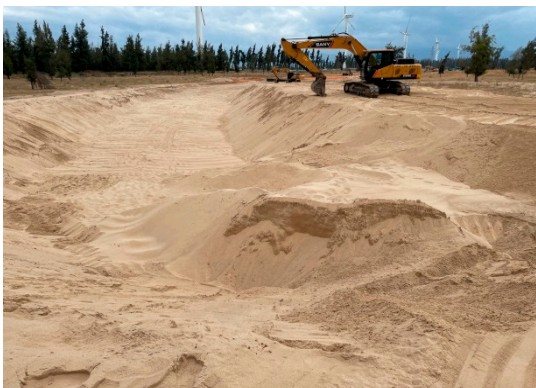
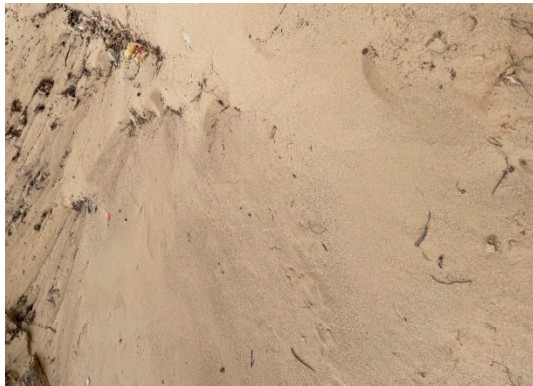

**Figure 1.** Engineering field.

Indoor soil tests were conducted on the sampled soil (Figure 2) according to the Geotechnical Test Regulations. The dry density of the sand sample was measured at 1.64 g/cm$^3$, as shown in Table 1. The sand exhibited minimal cohesion, and the basic physical parameters and particle gradation curves are illustrated in Figure 3.

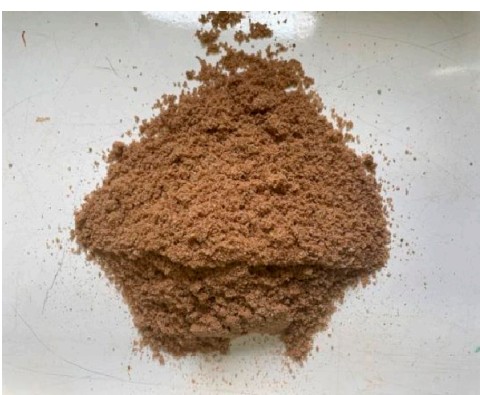

**Figure 2.** In situ sand sample.

**Table 1.** Sand property.

| Dry Density ($\rho$) | Specific Gravity ($G_s$) | Moisture Content (%) | Poriness | Cohesion ($C$) | Internal Friction Angle ($\varphi$) |
|---|---|---|---|---|---|
| 1.64 | 2.66 | 15 | 0.29 | 0 | 34 |

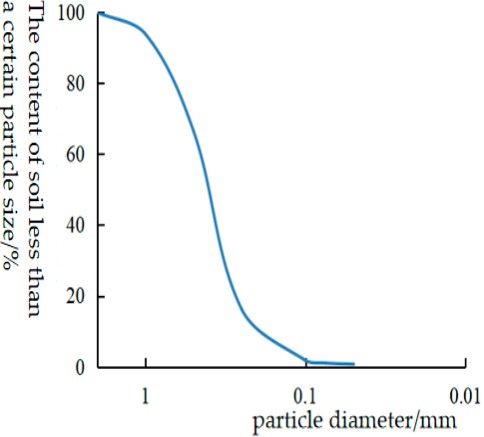

**Figure 3.** Grain composition.

## 2.2. Microbial Material

The microbial solidified strain used in the laboratory belongs to Bacillus and was purchased from Shanghai Preservation Biotechnology Center under the name ATCC11859. It was activated by freeze-dried powder in the biological laboratory into 500 mL liquid bacterial solution with OD600 = 2.0 (Figure 4).

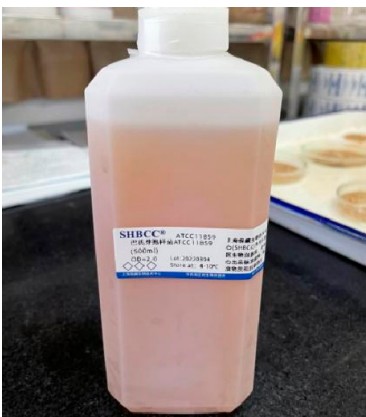

**Figure 4.** Bacterial stock solution.

The main components and contents of liquid medium are shown in Table 2. A total of 100 mL of the prepared liquid medium was subjected to sterilization by steam at 120 °C. Next, NaOH solution was added to adjust the pH value of the medium to 9.2. Subsequently, 20 mL, and it's concentration is 1 mol/L urea solution was added, and the cooled bacterial solution was inoculated into the medium at an 8% inoculum volume with a pipette gun. Following inoculation, the strains were incubated in a shaking bed at 30 °C with a shaking frequency of 180 r/m. After an incubation period of 27 h, the OD600 of the bacterial solution ranged from 0.6 to 0.8, as measured by a spectrophotometer, while the urease activity reached a range of 400~600 µs/cm, as measured by a conductivity meter.

**Table 2.** Main components and contents of liquid medium.

| Composition | Distilled Water | Bacteriological Peptone | Glucose | NaCl |
|:---:|:---:|:---:|:---:|:---:|
| **Content** | 1 L | 20 g/L | 20 g/L | 10 g/L |

## 2.3. Testing Apparatus

The test utilized the DH3818Y Static stress–strain data acquisition instrument manufactured by Donghua Company, which comprises 16 channels. The pore water pressure sensor used in the experiment was procured from Harbin Dayigangzhen Technology Co., Ltd., Harbin, China. The sensor, named the SDYH-DSPP-1 integrated miniature osmotic pressure sensor, is designed to measure negative pressure with a range of −50~50 kPa (piezoresistive type). It conforms to international standard sensors to monitor and measure the dynamic pore pressure changes in the slope rainfall model, as illustrated in Figure 5. The measured signal of the sensor represents the change in the pore water pressure by the voltage change rate, and the calibration coefficient is shown in Table 3.

The overall dimensions of the model box are shown in Figure 6, with a length, width and height of 500 mm, 500 mm and 200 mm, respectively. The box was constructed with three sides made of 10 mm thick toughened glass, while the bottom was a 5 mm thick steel plate. The model box was reinforced with steel plate columns at its four corners. Six circular drainage holes with a diameter of 4 mm were distributed at the bottom of the model, forming a 3 × 2 symmetrical pattern. The drainage holes should meet the basic requirements for proper drainage, and the liquid at the bottom should not be redundantly precipitated, which will affect the experimental results.

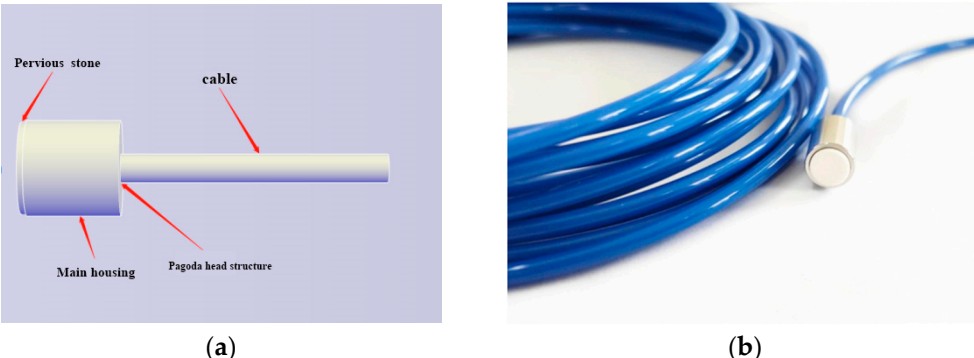

(**a**)  (**b**)

**Figure 5.** Stress–strain data acquisition instrument in figure: (**a**) sensor construction; (**b**) picture of real products.

**Table 3.** Sensor calibration coefficient.

| Sensor | Calibration Coefficient |
|---|---|
| 1 | 3298.62 kPa/V |
| 2 | 3316.59 kPa/V |
| 3 | 3501.72 kPa/V |
| 4 | 3207.78 kPa/V |
| 5 | 3279.31 kPa/V |

Based on the actual MICP curing condition, a specialized injection device was designed, which consisted of a pump, five nozzles, a timer and multiple hoses. The solution was vaporized by the surface spray method to increase the reaction surface area of the bacterial solution adhesion, the device model is shown in Figure 7. The pump used in the injection device has a maximum power of 45 W, and it can deliver a flow parameter of 4 L/min. The hoses were fixed after winding the axis for four turns, and all five sprinkler heads were situated on the same plane. The device can move left and right along the axis and rotate 360° to cover the required spraying surface area for the experiment. According to a certain number of spraying times and spraying volume, the fixed liquid, bacterial liquid and cementing liquid were sprayed vertically onto the slope, and the spraying device was positioned 300 mm away from the top of the slope. Unlike the single-channel reinforcement method, the reinforcement time can be reduced to less than 1 min, which greatly improves the reinforcement efficiency.

To simulate the scour effect, a homemade household double shower device with an adjustable tap was used to control the water flow, as shown in Figure 8. The test rainfall was determined according to the weighing method. The principle was to place a round mouth fitting with a diameter of 20 cm underneath the spraying device located 50 cm above the fitting. The rainfall was measured by a simple conversion, where every 30 g of water corresponds to 1 mm of rainfall.

### 2.4. Testing Program

According to the model box size, a 1:25 scale model slope M-1 with a height of 200 mm was prepared, and a 5 m high cutting slope of the coastal highway was simulated. The field sand sample with layered compaction (four layers) and sprinkling model was dried to control the moisture content at 20%. The slope gradient was set at 1:1. The basic parameters of the slope model are shown in Table 4. During the test, changes in pore pressure during the perfusion process and scouring action were measured by the embedded pore pressure sensor. The buried position of the sensor is illustrated in Figure 9. Sensors 1, 2 and 3 were vertically arranged along the depth direction, with a separation of 50 mm. Sensors 2, 4 and 5 were located at the same horizontal line, 100 mm below the top surface of the slope.

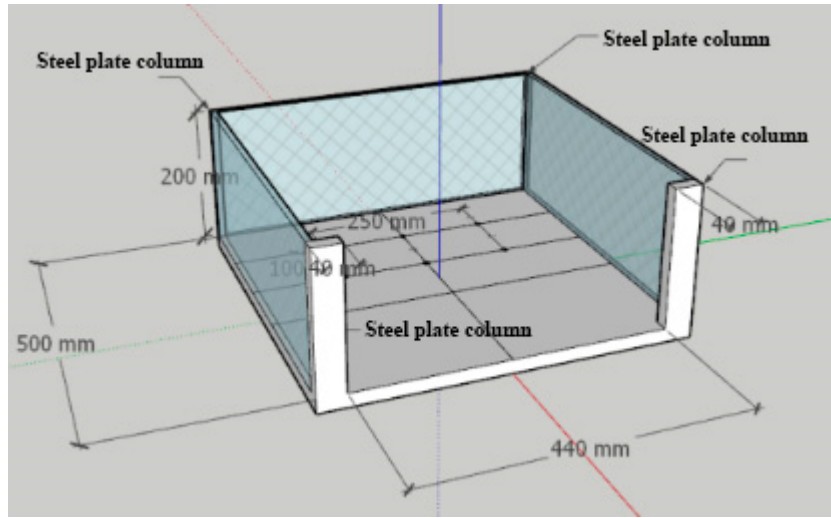

(**a**) 3D diagram of the model

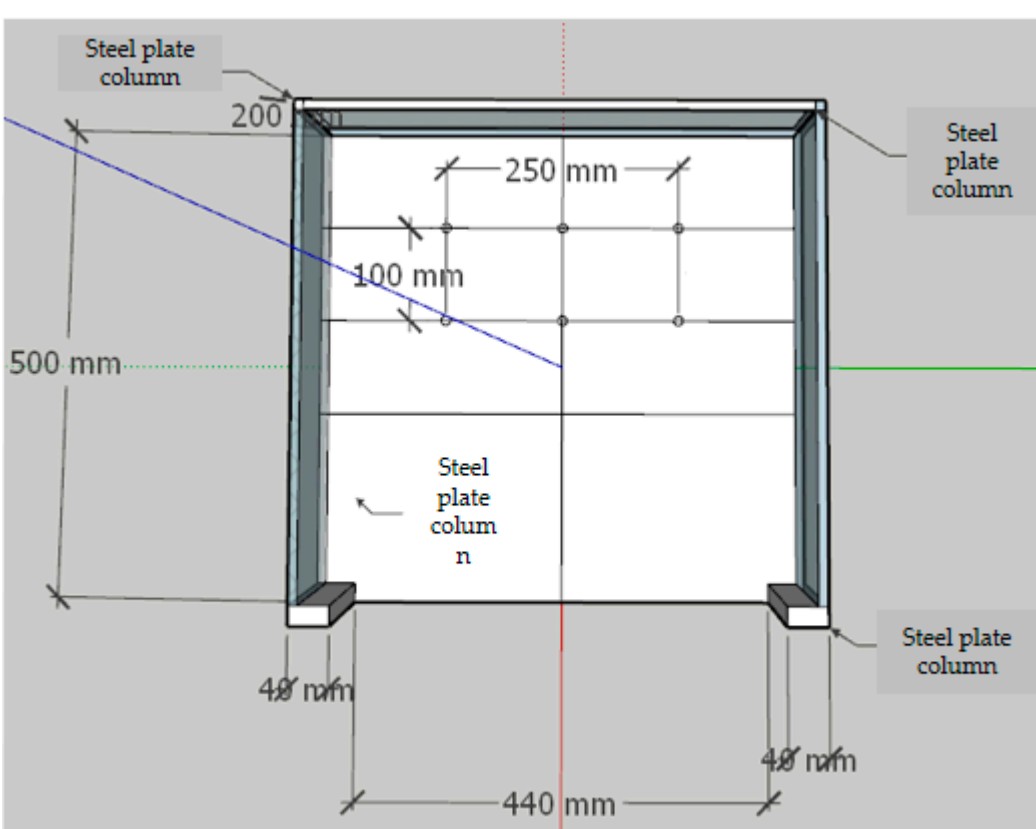

(**b**) Top view of the model

**Figure 6.** Model box size schematic.

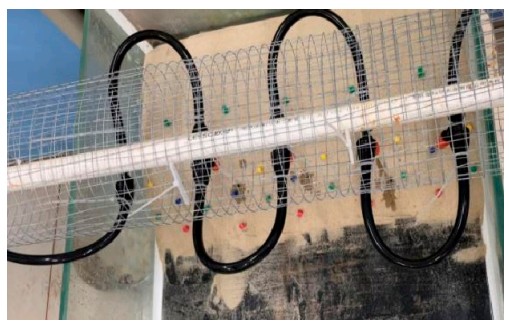
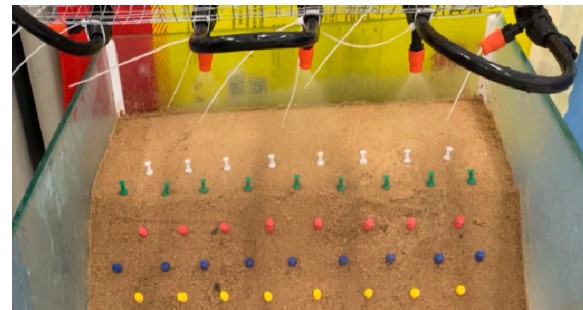

(**a**) The state before the experiment    (**b**) Experimental state

**Figure 7.** Sprinkler irrigation equipment.

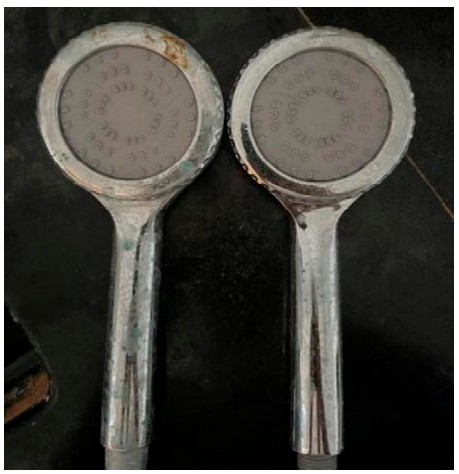

**Figure 8.** Double mounting device.

**Table 4.** M-1 basic parameters of slope model.

| Slope Base Length | Crest Length | Slope Height | Slope Gradient | Moisture Content | Reduced Scale |
| --- | --- | --- | --- | --- | --- |
| 300 mm | 100 mm | 200 mm | 1:1 | 20% | 1:25 |

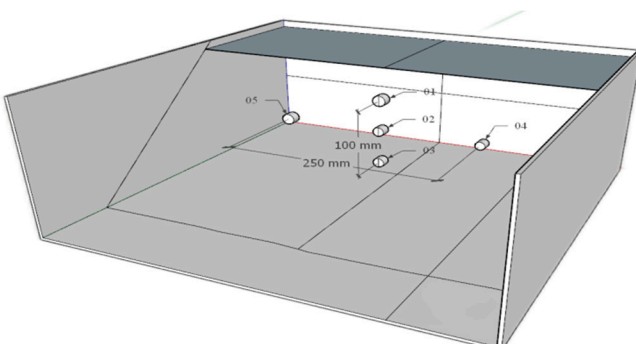

**Figure 9.** Internal sensor buried location diagram.

To facilitate the observation of displacement and deformation of the model under the action of scouring, displacement monitoring pushpins were arranged on the surface. These pushpins were spaced at 5 cm intervals along the transverse and slope directions. At the top of the slope, a row of displacement monitoring pushpins was arranged, precisely located at the leading edge of the slope. Four rows were arranged on the slope to construct the thumbtack surface monitoring point system. Each thumbtack was inserted 5 mm deep

into the soil to ensure that the model sand sample was minimally disturbed, as shown in Figure 10.

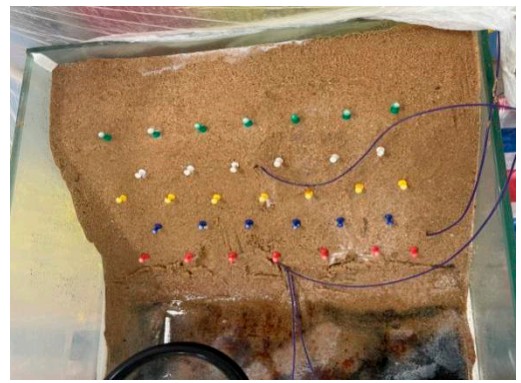 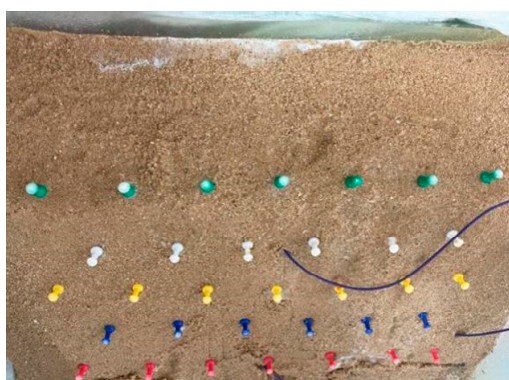

(**a**) First spray　　　　　　　　　　　　　(**b**) CaCo₃ is generated

**Figure 10.** Experimental model display.

The cementing solution was selected with a mix ratio of 1:0.5 ($CaCl_2$: urea). First, the $CaCl_2$ fixative solution with the same volume and concentration of 1 M was injected before the injection of bacterial solution. The injection process consisted of three sets of rounds, where the primary fixed solution + primary bacterial solution + primary cementing solution were injected into the sprinkler irrigation device in the order of fixed solution–bacterial solution–cementing solution, as shown in Table 5 for their respective volumes. The M-1 model underwent reinforcement through sprinkler irrigation in three rounds. The overall model is shown in Figure 11.

**Table 5.** Solution irrigation scheme.

| Stationary Liquid | Bacteria Solution | Consolidating Fluid | Time Interval between the Three Solutions |
| --- | --- | --- | --- |
| 1000 mL | 1000 mL | 2000 mL | 10 min |

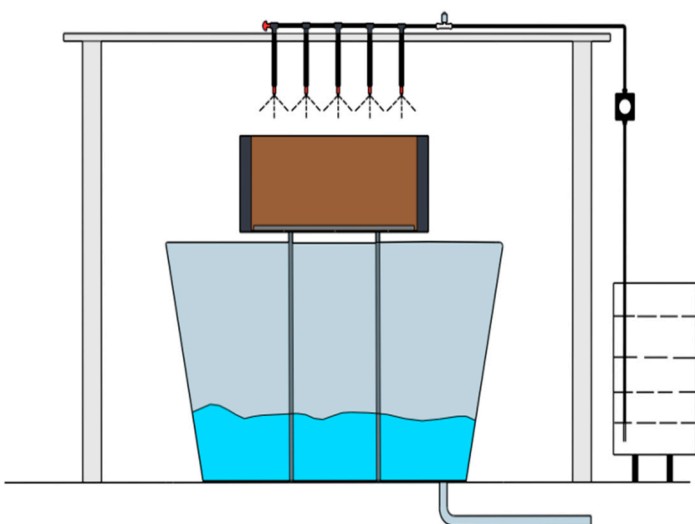

**Figure 11.** Overall model design.

## 3. Experimental Analysis

### 3.1. Pore Water Pressure

The looser the particles in the model, the greater the bacterial adsorption capacity. The sand sample with a smaller particle size will affect the change in the pore structure of the

sand model, with a reduced permeability coefficient. During the continuous reinforcement process, the permeability coefficient k weakens due to a gradual reduction in the pores of the upper sand sample [19]. As bacillus liquid and fixed liquid flow through the sand sample model, the pore water pressure in the pores will change with the seepage effect. In addition to surface runoff, a portion of the water flow direction in the sand sample also shows an infiltration effect [6,20]. After multiple rounds of reinforcement, the pores on the top surface, inclined surface and inside the sand sample model become filled and connected due to the formation of calcium carbonate particles. The pore water pressure sensor can be used to explore the changes in pore pressure during reinforcement and pore characteristics after multiple rounds of reinforcement. He Xiaoying, Li Dong et al. [21] equipped the laboratory with debris flow two-phase bodies to observe the change rule of pore water in debris flow sediment reinforced by MICP. They found that calcium carbonate generated by MICP solidification has a tendency to retain pore water pressure. In drainage consolidation tests, it was observed that when the sediment porosity was low, it was difficult to drain and consolidate. Li Xian proposed a theoretical formula for the grouting capacity of sand with different permeability coefficients of MICP, which provides a reference for whether the soil is suitable for reinforcement by the MICP method [5].

The pore water pressure test was divided into two stages. In the first stage, the M-1 model was injected with several different volumes of fixed liquid and bacterial liquid, and the change rule of the internal pore water pressure was measured by the embedded pore water pressure sensor inside the slope. In the second stage, a self-made rainfall device was used to simulate a certain rainfall intensity and explore the change process of pore water pressure from the initial moisture content to the saturation point of the entire sandy slope until its internal water pressure law becomes unstable [22].

As observed from Figure 12a, the 02 sensor in the middle and the sensors on both sides of the same horizontal plane of the solution infiltration channel were concentrated in the middle of the M-1 model during the first reinforcement, and the signals in the middle channel were very clear and distinct, while the signals at both ends of 02 were not as evident, and the pore pressure variation law was basically similar. Moving on to signal (b), during the second reinforcement, the seepage channel was concentrated on the right side of the model, and the secondary wave crest on the left side was not as pronounced. The variation in the time–pore pressure in the horizontal direction of the third round of reinforcement (c) indicates that the seepage channel is located on the left of the center of the model. The test results show that after three rounds of perfusion reinforcement, the changes in pore pressure in the horizontal plane, specifically 10 cm away from the top surface, exhibit clear sensitivity to the seepage layer. At this depth, the calcium carbonate products are comparatively lower, with an ordinary filling effect, and the curing effect is not particularly noticeable at this depth.

From Figure 13, it is evident that the pore pressure undergoes changes from top to bottom during the cementing process. Sensors 02 and 03 showed a certain time lag compared with 01, and the size effect was more obvious. First, to verify the possible interaction between the bacterial solution and the cementing solution on pore pressure changes, a test was conducted where 400 mL of bacterial solution was injected at 280 s for 20 s. The test results show that the rising rate of the shallow sensor 01 is the fastest after the mixing of the cementing liquid and bacterial liquid, signifying an obvious reaction of the calcium carbonate surface. However, after the rise in the pore pressure at the bottom, the secondary wave crest changes minimally, and the pore pressure dissipates for an extended period. As a result, the pore pressure change rate of 01 at the secondary wave peak is not as high as that of 02 and 03 at the bottom. The results suggest that the reactants gradually reduce, and the formation of calcium carbonate is more obvious in the shallow reaction, hindering the infiltration of the solution. However, in Figure 13b, after 01 began to rise, the pore pressure remained constant at 0.33 kPa for about 500 s. This indicates that, in the shallow depth range, due to the shallow reaction of calcium carbonate, after filling the sand pores, the pores slowly decrease until completely filled. In terms of the pore

pressure law, the change rate of the pore pressure weakens and remains unchanged for a certain period, approximately forming a kind of "calcium carbonate film" that stabilizes the pore pressure. However, sensors 02 and 03 below can still perceive the change in the pore pressure after solution infiltration. The pore pressure rises quickly within a certain period, and the process of pore pressure dissipation also indicates a gradual decrease in the pore water pressure in the model, reflecting the effect of infiltration. The accumulation of calcium carbonate in the surface pore of the model greatly affects the changes in the surface pore pressure.

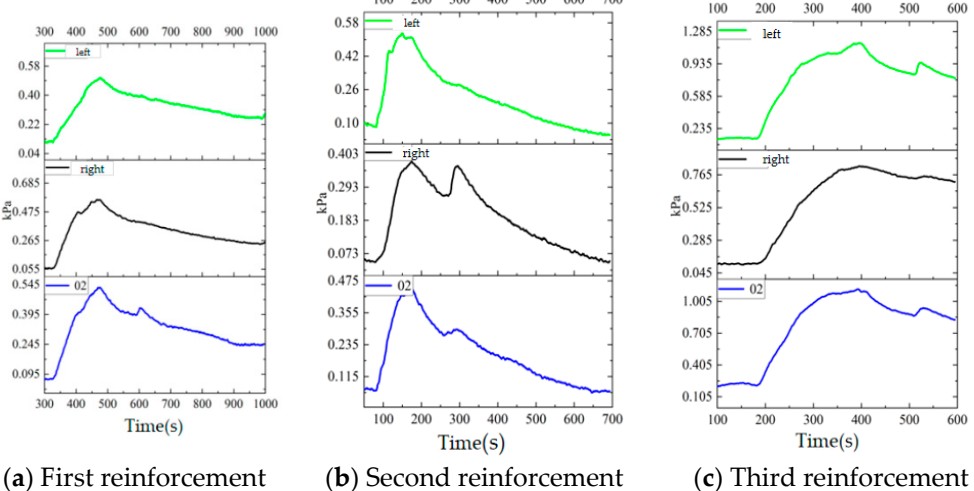

(**a**) First reinforcement    (**b**) Second reinforcement    (**c**) Third reinforcement

**Figure 12.** Horizontal direction of the first, second, third wheel reinforcement pore pressure changes.

Figure 14a,b, respectively, show the time–pore pressure variation in the bacterial and cementing fluids in the third round of reinforcement. The continuous changes in the pore pressure in this cycle reaffirm the plugging effect of the calcium carbonate "film" formed in the middle pores of the upper part of the model. The previous stone of sensor 1 is blocked by the products near the measuring point of sensor 1, and the stabilization of the calcium carbonate around the sensor results in a minimal pore pressure change rate, showing a straight line on the graph. On the other hand, the deep sensor is characterized by a dynamic equilibrium process from shallow to deep, indicating that the deep solidification is not significant, and the lower pore cementation effect is poor. Pore water during grouting is represented as a dynamic infiltration process, in which the pore pressure changes in 01 in the three groups are mapped. As shown in Figure 15, after irrigating three times with bacterial and cementing fluids, the pore water pressure at 01 gradually decreases. Compared with the second irrigation, the decrease in the first irrigation is the most obvious, which can be attributed to the large amount of calcium carbonate generated in the first two irrigations, filling most of the voids and leading to a minor change in the graph between the third irrigation and the second irrigation. This suggests that the downward infiltration of bacterial and cementing fluids in the third irrigation is affected by the substantial amount of calcium carbonate generated in the previous two irrigations, confirming the conclusion that the deep curing effect is not significant.

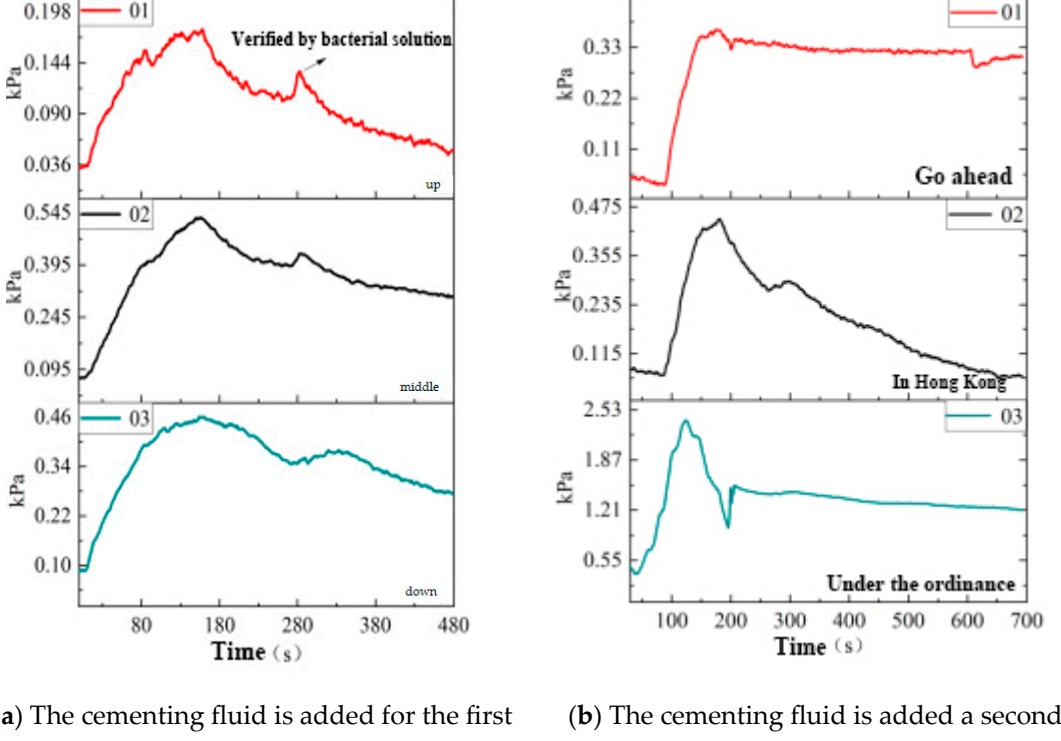

(**a**) The cementing fluid is added for the first time

(**b**) The cementing fluid is added a second time

**Figure 13.** Pore pressure changes in the first and second rounds of cementing fluid.

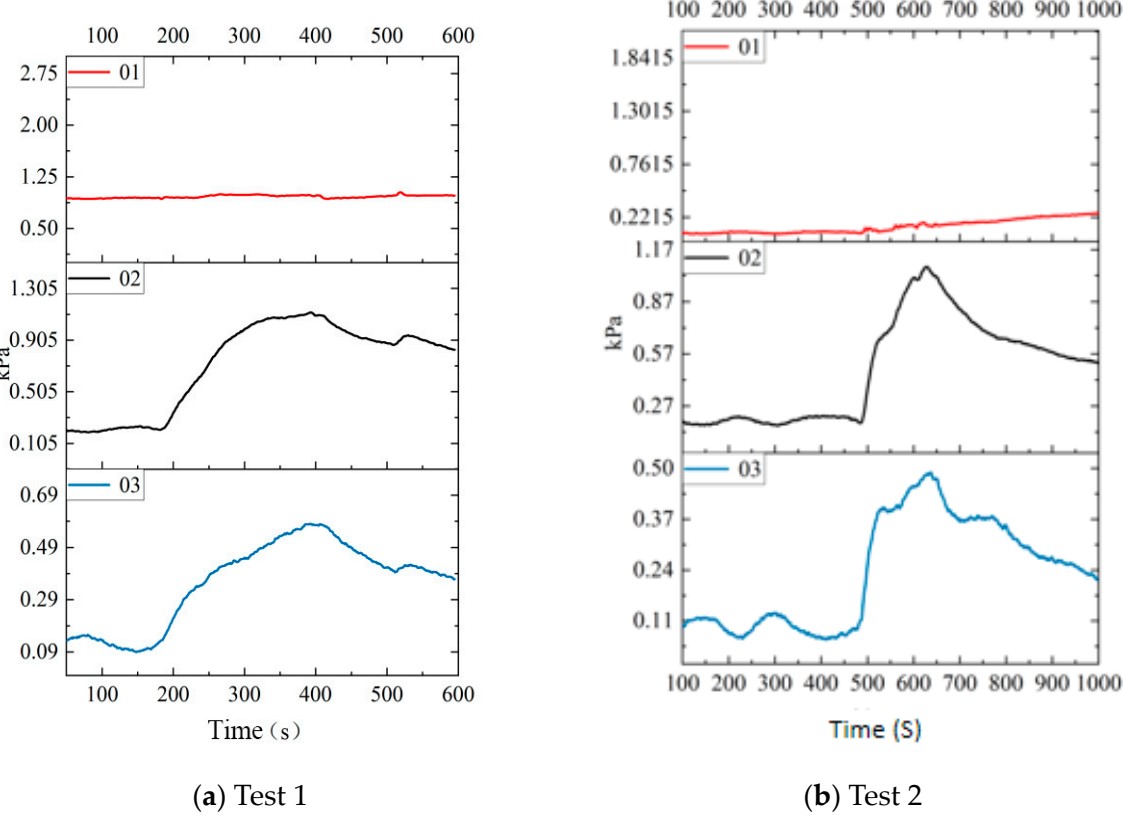

(**a**) Test 1

(**b**) Test 2

**Figure 14.** Pore pressure changes in the third round of reinforcement when bacteria liquid and cementing liquid pass through.

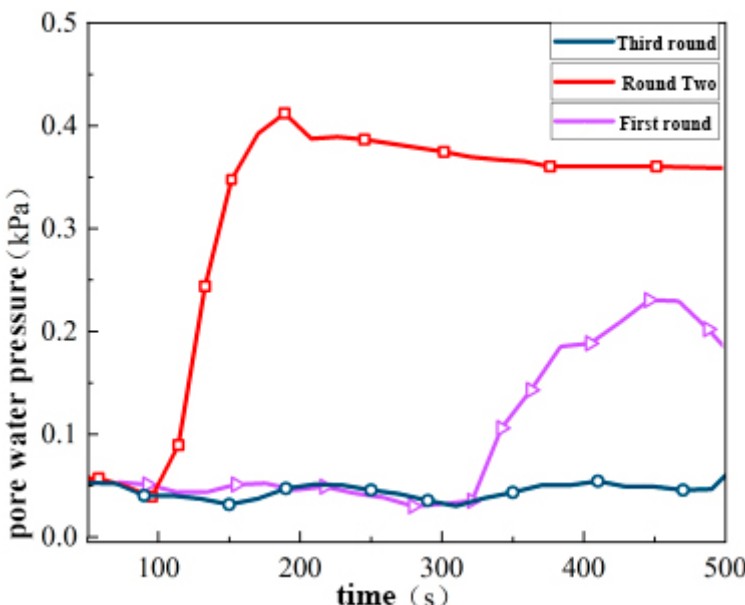

**Figure 15.** Changes in 01 time–pore pressure in three rounds.

By idealizing the slope model, the failure of the model can be divided into three stages under the short-duration and high-intensity rainfall condition. The overall time–pore pressure variation diagram for the 01–05 sensors and the time–pore pressure variation diagram for different time periods are shown in the figure below.

According to the analysis of Figures 16–20, the first stage is the infiltration stage. Based on the pore pressure changes in this stage, the pore pressure change rate of 01 is close to zero due to the existence of a calcium carbonate "diaphragm" in the pores of the sand body, as characterized by sensor 1. Solution percolation exists in the M-1 model. When the pore pressure drops in No. 02 of the upper part, the solution percolates downward, while the pore pressure rises in No. 03, exhibiting a top–down seepage change feature. The second stage is the rainfall stage. From 872 s, due to the surface solidification, the cementation of the slope surface and the top area Is obvious. The rainfall Is mostly transformed into runoff flowing down the slope surface, and the slope surface enters a transient saturation state. However, the cementation of the calcium carbonate on the surface limits the infiltration of a large amount of water, and the pore pressure of 02 and 03 does not change significantly. Due to the increase in water content, the decrease in basement suction leads to a decline in slope stability. When the slope begins to slip at 109 s, the basement suction in the soil becomes zero, and the slope model starts to slide and fail. After 160 s, the rainfall ends. The third stage is the sliding unloading stage, starting from 1033 s. After the rainfall, the overlying pressure continues to unload, and the slope is damaged in stages, causing the pore pressure to decrease in a stepped way. The pore pressure deformation law of the slope without MICP reinforcement should be a broken line of slow decline after failure. However, after MICP reinforcement, the creep of the S-3 slope is very slow after the rainfall stops. Due to the cementation in the sand body, the pore pressure in the slope discharges in a stepped manner, and the basement suction undergoes a process of repeated rise and then decline. After MICP reinforcement, the slope can maintain certain stability after failure and prolong the creep time.

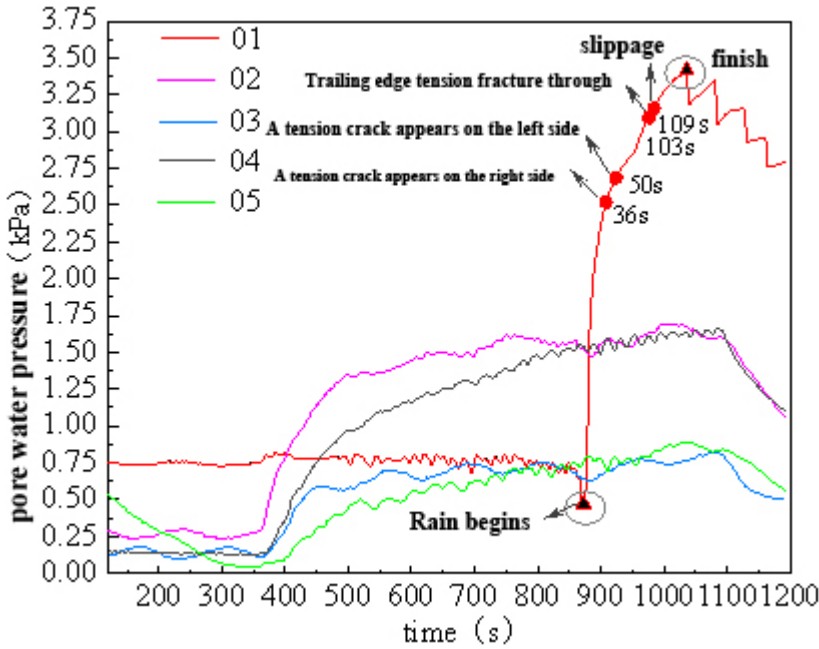

**Figure 16.** Variation in 01–05 pore pressure during rainfall.

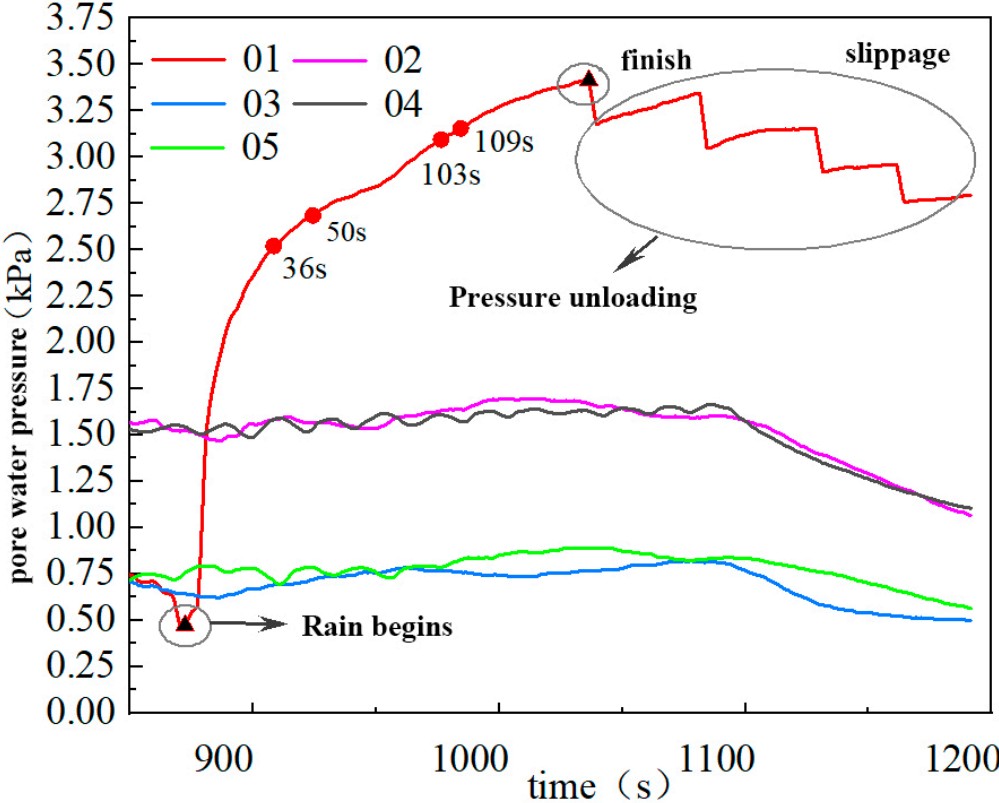

**Figure 17.** The 01–05 sensors rainfall duration–pore pressure change from 850–1200 s.

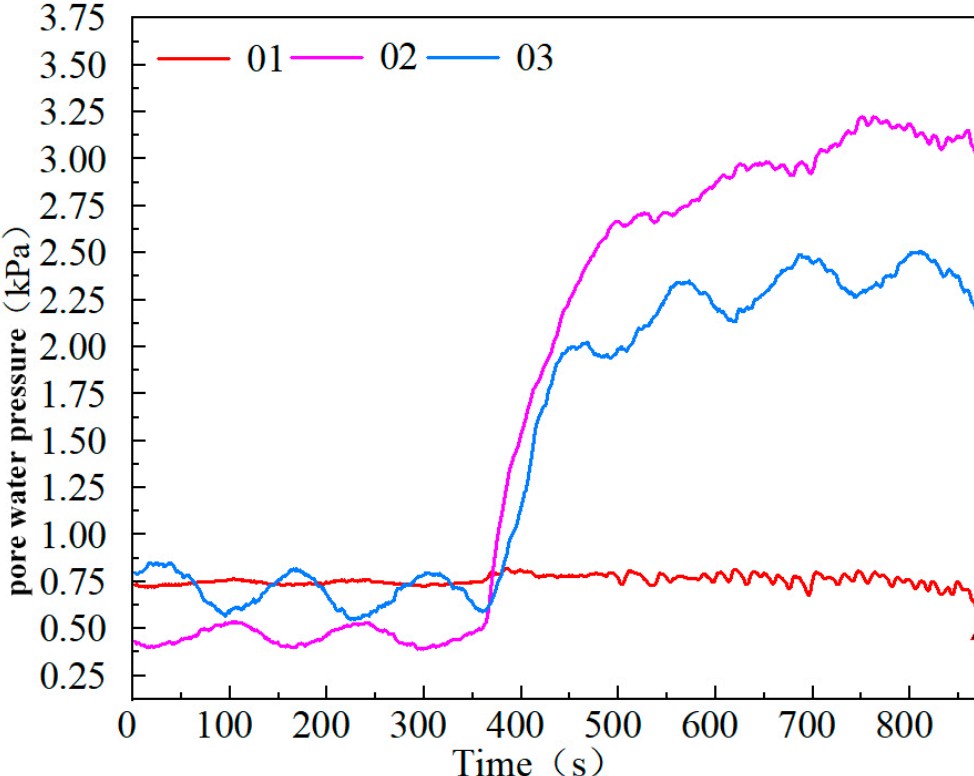

**Figure 18.** The 01–03 pore pressure changes when the sensor is 0–800 s.

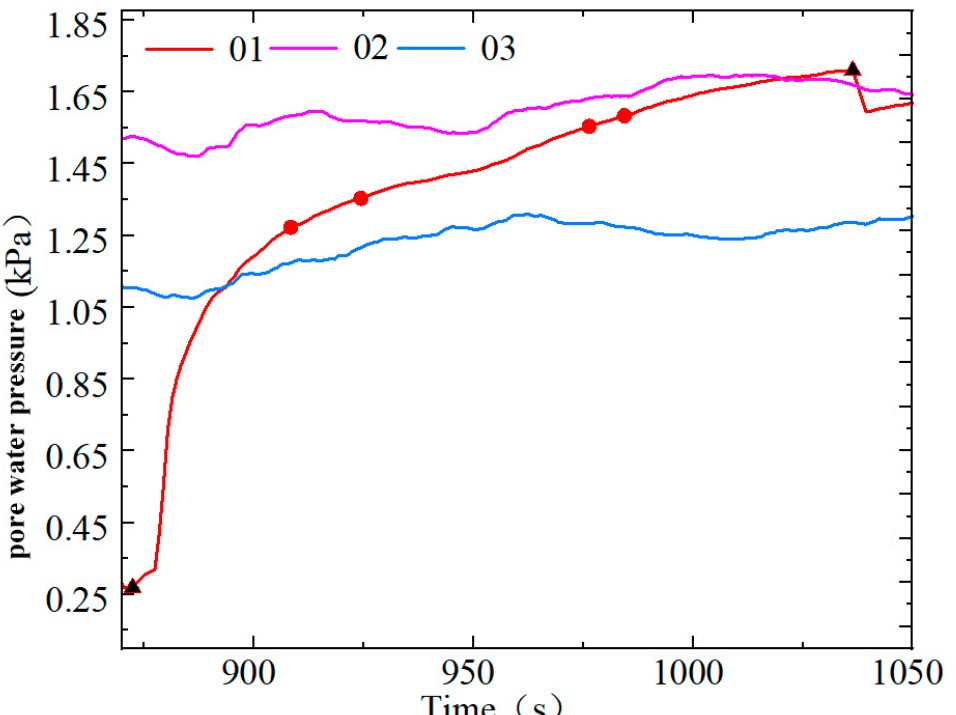

**Figure 19.** The 01–03 sensors pore pressure change from 800 to 1050 s.

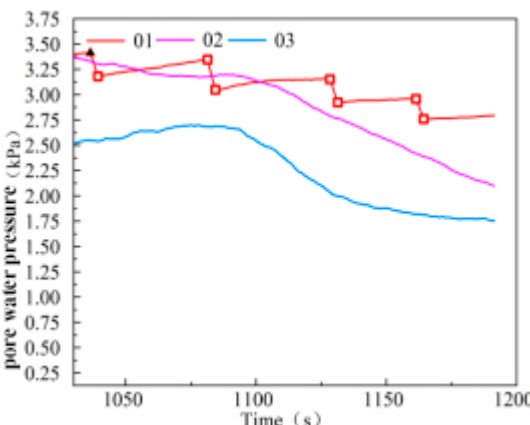

**Figure 20.** The 01–03 pore pressure changes in the sensor at 1050–1200 s.

### 3.2. Fluid Slip Pattern

After three rounds of reinforcement on the slope of the M-1 model, a simulation of the short-duration high-intensity scour action was conducted, and the flow rate of the scour device was measured. According to the regulations of the national meteorological department, the rainfall of 23 mm reached the heavy rain level within 12 h (15–29.9 mm), and the rainfall intensity was 23 mm/min, exceeding the local average daily rainfall standard in Dongfang City. The damage morphology of M-1 was observed after being scoured.

At 150 s, M-1 fails under the rainfall simulation, and the slope experiences a prolonged period from the beginning of erosion to sliding failure. After the erosion of the rainfall intensity, the failure mode manifests as the overall sliding failure of the slope. After 36 s, the fracture begins to develop on the right side of the M-1 slope top, with a diagonal fracture length of 4 cm and a width of 2 mm. After 50 s, the left oblique crack starts to develop, measuring 6 cm in length and 2 mm in width. The sand on the slope surface remains intact despite the entrainment runoff, making it resistant to loss. Approximately 103 s into the process, the oblique cracks on both sides connect, and the tension cracks appear at the rear edge of the slope top, with a width of the cracks of 3 mm. These cracks are distributed in a circular arc. The green thumbtack band on the slope top moves downwards. The left side of the slope surface is missing, and the red pushpin belt at the slope angle begins to deform. The slope foot experiences unloading. The moisture content of the lower soil is close to saturation, but the slope body integrity is well-maintained. Refer to Figure 21 for further details.

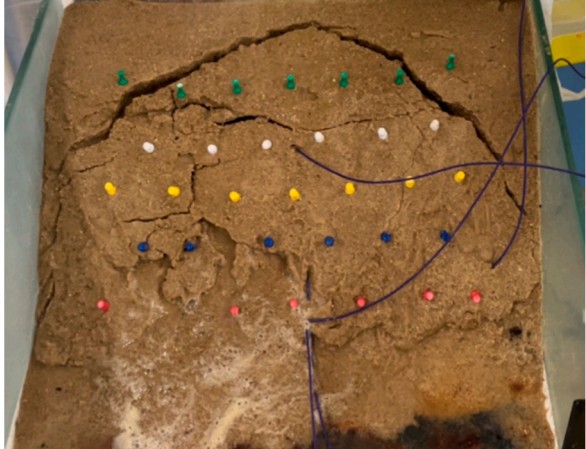

**Figure 21.** Integral slip.

At 115 s, the scour effect leads to an increase in the internal water content of the soil, causing a decrease in effective stress and the anti-sliding force of the soil. Consequently, the slope surface of the model begins to slide downward as a whole. The development of tension cracks induces rainfall convergence, and the infiltration of rainwater further raises the water content of the model, which accelerates the sliding of the slope. Despite some surface damage, the model maintains good overall integrity. The failure characteristics of the model are different from those of the model without MICP reinforcement. The sliding surface inside the slope can be seen from the lower M-1 side view, and it can be inferred that the sliding surface is located at 4–5 cm depth. For further details, refer to Figure 22.

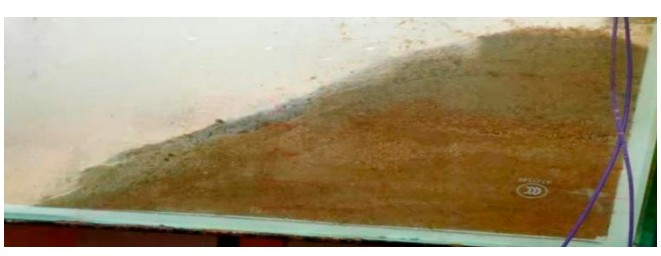
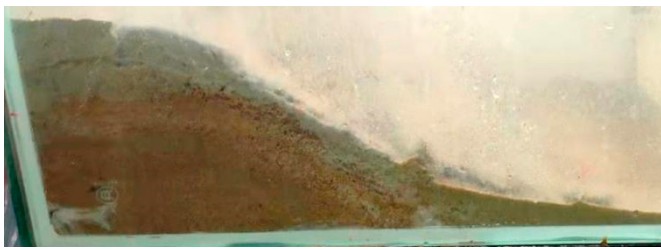

(**a**) Side view 1  (**b**) Side view 2

**Figure 22.** M-1 model slippage side view.

When the simulated rainfall effect is stopped, the slope model morphology is divided from the front and top surfaces due to the development of cracks. The surface sand of the slope block remains intact, and the segmented blocks have a thickness of about 3–4 cm. After 150 s of simulated scour, the model experiences damage. The failure characteristics of the MICP-solidified slope model M-1 are manifested as fractional sliding. Due to sand loss, the front edge of the model exhibits an unloading effect and the largest deformation. After the slope angle slip and accumulation, the upper slope continues to slide and deform slowly. The damage integrity of the model after MICP action remains good in the short-duration and high-intensity simulation of rainfall, and the MICP spray irrigation curing technology can effectively resist surface erosion. For further details, refer to Figure 23.

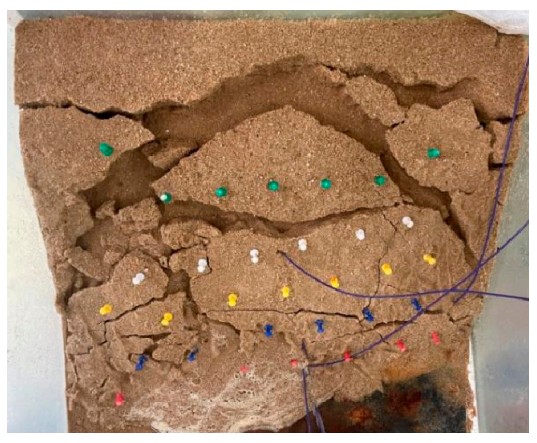
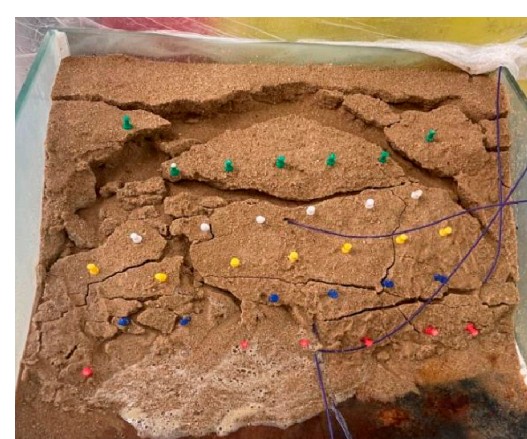

(**a**) Front view  (**b**) Top view

**Figure 23.** View after destruction.

## 4. Conclusions

(1) In the M-1 model, after two rounds of reinforcement, a calcium carbonate "film" was formed in the shallow pores of the slope, which reduced the pore pressure change rate during the infiltration of the solution. The model first reacts to produce calcium

carbonate in the shallow layer, leading to a solidification of the shallow layer and a gradual decrease in the space between the sand particles. The lower part of the model demonstrates the process of solution infiltration, and the secondary wave crest rises slowly, indicating that the product prevents solution infiltration. The pore pressure of the sensor at the same horizontal depth follows the same change law. However, below 10 cm of the top surface, there are fewer products, with a larger pore pressure change rate and general curing effect.

(2) The S-3 slope, strengthened by three sets of turns, failed after 150 s. The surface blocks of the slope were divided due to the development of cracks. The thickness of the divided blocks was about 3–4 cm. After 109 s of rainfall, the moisture content of the soil increased, with a decrease in the effective stress and anti-sliding force of the soil and the slope surface of the model as a whole. The development of tension cracks induces rainfall convergence and accelerates slope slip. Despite some surface damage, the model maintained good overall integrity. The failure characteristics of the slope are manifested as fractional sliding. Due to the loss of sand, the unloading effect of the model front is obvious, and the deformation is the largest. At the onset of the slip, the suction of the base became zero, and the failure form shows an overall slip with subsequent creep unloading.

(3) After MICP reinforcement, the creep of the M-1 slope model was very slow. The cementation in the sand body led to the discharge of the pore pressure in the slope in a stepped manner, and the basement suction underwent a process of rising and then falling repeatedly. The MICP slope can maintain a certain integrity after failure, prolong the creep time, improve the water stability and anti-erosion capacity of the MICP slope under the action of heavy rainfall and enhance the safety factor of the slope.

**Author Contributions:** Conceptualization, J.H. and Y.H.; methodology, M.H.; software, Z.X.; validation, J.S.; formal analysis, M.H.; investigation, Y.H.; resources, Z.X.; data curation, J.S.; writing—original draft preparation, Y.Z.; writing—review and editing, Y.H.; visualization, M.H.; supervision, Y.Z.; project administration, J.H.; funding acquisition, Y.Z. All authors have read and agreed to the published version of the manuscript.

**Funding:** This research was supported by Hainan Provincial Natural Science Foundation of China (NO. 521RC1040, NO. 522CXTD510) and Major Science and Technology Program Projects of Hainan Province (NO. ZDKJ2021024).

**Institutional Review Board Statement:** Not applicable.

**Informed Consent Statement:** Not applicable.

**Data Availability Statement:** Not applicable.

**Conflicts of Interest:** The authors declare no conflict of interest.

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
