# Peer review of "Experimental Study on Pore Pressure Variation and Erosion Stability of Sandy Slope Model under Microbially Induced Carbonate Precipitation"

_sustainability, doi:10.3390/su151612650_

Round 1

Reviewer 1 Report

This paper applied the microbial mortar generated by microbial grouting to the slope reinforcement, and obtained better microbial grouting data through the grouting test. The pore water pressure monitoring is carried out on the slope model during MICP solidification and rainfall scour. Combined with the model pore water pressure and flow slip failure pattern, dynamic analysis was carried out. In this study, a new method of MICP remediation techniques is proposed, which provides a new test basis for the application of MICP technology in practical engineering. This is a useful paper and is well-written containing some interesting results. It is acceptable for publication following consideration and response to the following points.

(1) Lines 149-152: Table 1 and Figure 3 should be clearly indicated.

(2) Figure 6 does not show the specific distribution of drainage holes.

(3) The name of Figure 7 is not given.

(4) Line 237: the “and the slope slope is 1:1” should be “and the slope gradient is 1:1”.

(5) Line 304: the “Figure (a)” should be “Figure 12(a)”.

(6) Figures 16-20: Different linear or different shapes should be used in the Figures. It is not obvious to distinguish only by color.

(7) 3.1 Fluid-slip pattern: Could the displacement quantitative analysis be added to be consistent with the pore water pressure response, which would be more convincing.

no

Author Response

1-5,7 it's been changed

6,These pictures are from professional instruments. I have tried many times and can't change them.

Reviewer 2 Report

Dear Editor Journal of Sustainability

The article entitled “Experimental study on pore pressure variation and erosion stability of sandy slope model under MICP”, have useful scientific information. But, paper in present format require to minor revisions. Specific remarks are as follow:  

1- Important results should be expressed quantitatively in abstract.

2- In terms of grammar, paper should be re written again.

3- What is the value of relative density in sand slope? How relative density have been controlled in top and down of slope? It should described in text? 

4- Permeability value of sand is important in MICP method? Also, curing time? It should be described in text.  

Best regards

Dear Editor Journal of Sustainability

In terms of grammar, paper should be re written again

Author Response

1 and 2 have been modified

3,4 is indeed we did not consider, has been described in side
